# Knowledge of COVID-19 and preventive behaviors among waiters working in food and drinking establishments in Southwest Ethiopia

Qaro Qanche[1], Adane Asefa[1]*, Tadesse Nigussie[1], Shewangizaw Hailemariam[2], Tadesse Duguma[3]

1 Department of Public Health, College of Health Science, Mizan-Tepi University, Mizan-Aman, Ethiopia,
2 Department of Midwifery, College of Health Science, Mizan-Tepi University, Mizan-Aman, Ethiopia,
3 Department of Medical Laboratory, College of Health Science, Mizan-Tepi University, Mizan-Aman, Ethiopia

* adane779@gmail.com

## Abstract

### Background

Waiters working in different food and drinking establishments have a higher risk of contracting COVID-19 and transmitting the infection to others because they interact with many people. Most COVID-19 related studies in Ethiopia mainly focused on the general population, whereas, this study aimed to assess the knowledge of COVID-19 and preventive behaviors among waiters in Southwest Ethiopia.

### Methods

A cross-sectional study was conducted from June 1 to June 15, 2020, among waiters working in food and drinking establishments found in Mizan-Aman, Jemu, and Masha towns in Southwest Ethiopia. A total of 422 waiters were selected using a simple random sampling technique, and the data were collected through face-to-face interviews using a structured questionnaire. The data were entered into Epi-data manager version 4.0.2 and analyzed using SPSS version 22. Multivariable binary logistic regression analysis was carried out to identify predictors of good preventive behaviors at a p-value of less than 0.05.

### Results

Four hundred and sixteen respondents participated in this study, with a response rate of 98.6%. A significant proportion of participants know the cause, route of transmission, symptoms, and prevention methods of COVID-19 virus. However, very few (21.2%) had good preventive behaviors. The study showed that good preventive behavior was positively associated with female sex (AOR = 2.33, 95% CI: 1.38–3.94), higher schooling (AOR = 0.39, 95% CI: 0.17–0.88), high-risk perception (AOR = 2.26, 95% CI: 1.51–4.32), and high perceived self-efficacy (AOR = 1.1.75, 95% CI: 1.05–2.90).

**Data Availability Statement:** All relevant data are within the manuscript and its Supporting Information files.

**Funding:** The author(s) received no specific funding for this work.

**Competing interests:** The authors have declared that no competing interests exist.

**Abbreviations:** AOR, Adjusted Odd Ratio; CI, Confidence Intervals; SPSS, Statistical Package for Social Scientists; WHO, World Health Organization.

## Conclusions

A significant proportion of waiters know common symptoms of COVID 19, route of transmission, and its prevention methods. However, the preventive behavior was very low. Thus, all concerned bodies working on the prevention and control of COVID-19 should give attention to this population group to enhance compliance with recommended preventive behaviors.

## Introduction

Coronavirus disease 2019 (COVID-19) is an infectious disease caused by the novel severe acute respiratory syndrome coronavirus 2 (SARS-CoV-2). It was first discovered in the Hubei province of China in December 2019 [1] and on 11[th] March 2020, the World Health Organization (WHO) announced the outbreak of COVID-19 as a global pandemic due to the rapid increase in the number of cases outside China [2]. In Ethiopia, the first case of COVID-19 was reported on 13[th] March 2020 [3].

COVID-19 virus is transmitted from person-to-person through respiratory droplets, direct contact with an infected individual, or indirect contact with a surface or object that is contaminated with respiratory secretions [4]. Although not common, coronaviruses can also be transmitted from animals to humans [5]. During the early phase of the pandemic, the WHO suspected the zoonotic source of the virus and recommended precautionary measures to reduce the risk of transmission of emerging pathogens from animals to humans [6]. The disease is clinically presented with fever, cough, difficulty breathing, and other flu-like signs and symptoms including runny and stuffy nose, sneezing, and sore throat. While most people with COVID-19 develop only mild to moderate (81%) disease, approximately 15% develop severe disease that requires oxygen support, and 5% have critical disease with complications [7–9].

Currently, there is no effective treatment and vaccine for the virus. However, active case finding and isolation, quarantine, frequent handwashing with water and soap or alcohol-based sanitizers, social distancing, avoiding travel/travel restrictions, use of facemasks, and avoiding public gatherings are the measures of choice to prevent and mitigate the effect of the pandemic [10–14]. However, the effectiveness of such measures depends on public awareness and strict obedience to those recommendations.

Food and drinking establishments are potential hotspots for COVID-19 spread because many people share food, talk loudly, and drink alcohol in enclosed spaces [15]. The high interactions between guests and staff could lead to high transmission rates. Furthermore, the transmissibility of COVID-19 virus from asymptomatic patients could lead to a higher probability of work-related transmission as people with mild or no symptoms could continue to work or travel [16]. Thus, waiters working in different food and drinking establishments have a higher risk of contracting COVID-19 or easily spread the virus in the community.

Although waiters are at a higher risk of contracting and spreading the infection [16], information regarding their level of knowledge of COVID-19 and preventive behaviors is scarce. Most studies conducted so far have focused on the general public ignoring this vulnerable population group [17–19]. Therefore, the current study was intended to assess the knowledge of COVID-19 and preventive behaviors of waiters in Southwest Ethiopia.

## Methods and materials

### Ethical statements

Ethical approval was obtained from Mizan-Tepi University Institutional Review Board (IRB) before the commencement of the study. Written informed consent was obtained from all

participants after explaining the study's purpose, risks, and benefits. Moreover, participants were assured the participation is entirely voluntary and personal information is not disclosed to third parties.

## Study design and area

A cross-sectional study was conducted among waiters working in hotels, restaurants, cafeterias, and bars found in Mizan-Aman, Jemu, and Masha towns in Southwest Ethiopia from June 1 to 15, 2020. Mizan-Aman, Jemu, and Masha are the administrative centers of Bench-Sheko, West-Omo, and Sheka zones, respectively. Mizan-Aman town is located 585 km from Addis Ababa, while Jemu and Masha are found at 625 km and 6132 km from Addis Ababa, respectively. The areas are commonly known by gold mining and cash crop production, such as coffee and different spices. As a result, there are high social mobilities in the areas that make a conducive environment for the spread of the COVID-19 virus.

## Population

The source population was all waiters working in food and drinking establishments that were found in Mizan-Aman, Sheka, and Masha towns. Waiters on duties in the selected hotels, restaurants, cafeterias, and bars during the data collection period were randomly selected for interview.

## Sample size and sampling procedure

The sample size was calculated using a formula for single population proportion by considering the following assumptions: 5% margin of error, 95%confidence level, and expected proportion of waiters with good preventive behavior to be 50%. A proportion of 50% was taken because there were no previous similar studies in Ethiopia. After adding a 10% contingency for non-response, the final sample size was determined to be 422.

A simple random sampling technique was used to select the study participants. First, lists of all food and drinking establishments found in the three towns were obtained from the respective town administration. Then, the sample size was proportionally allocated to each town depending on the total number of food and drinking establishments in each town. Finally, simple random sampling technique was use to select establishments from each town using computer generated random numbers. At establishment level, one waiter per establishment was selected for interviews. If more than one eligible individual presented in selected establishments, one person was randomly selected using the lottery method. All waiters in an establishment were listed on separate slips of paper of the same size and shape. Then, papers were folded carefully, and finally, a blindfold selection was made.

## Measures

The data were collected through face-to-face interviews using a pretested structured questionnaire. The questionnaire was developed from related studies and guidelines [20–23]. The tool consisted of five parts: participants' characteristics (age, sex, marital status, religion, occupation, ethnicity, and the number of people living in the home), knowledge of COVID-19, perceived self-efficacy regarding prevention measures, COVID-19 preventive behaviors, and risk perception regarding COVID-19.

COVID-19 preventive behavior was measured using 10 items. Respondents were asked to rate how often they had been practicing the preventive measures recommended by the WHO during the pandemic on five-point scales: (1) never, (2) rarely, (3) sometimes, (4) frequently,

and (5) always. During analysis, each item was recoded into "not practiced or inadequate practice" if subjects scored never, rarely, or sometimes and into "adequate practice" if scored frequently or always. Finally, respondents who scored adequate practice for at least 60% of the items were considered as having "good preventive behavior"; otherwise "poor preventive behavior". A 60% cutoff point was achieved based on a study done in Iran [24].

Knowledge of COVID-19 (etiology, mode of transmission, symptoms, prevention methods, and treatment or vaccine) was measured using 15 items that were answered on" yes", "no" or "I don't know" responses. The correct answers were coded with 1 and the incorrect answers were coded with 0. Finally, participants who scored ≤ 59% were categorized as having "poor knowledge", 60%-79% as "moderate knowledge", and ≥80% as "good knowledge" of COVID-19 based on Bloom's cut-off point [25].

Self-efficacy to practice commonly recommended COVID-19 prevention methods was measured using 4 items that were responded on a five-point scale: (1) certainly not, (2) probably not, (3) perhaps not to perhaps yes, (4) probably yes, and (5) most certainly. The respondents were asked if they were able to carry out the recommended measures. A mean score was computed and a score at mean or less indicates low self-efficacy, while a score above the mean indicates high self-efficacy.

Risk perception regarding COVID-19 was measured using 12 items of five Likert scale: (1) strongly disagree, (2) disagree, (3) neutral, (4) agree, and (5) strongly agree. The items were stated in a way that a higher value indicates higher risk perception. The cumulative risk perception score (range 12–60) was computed [22]. Based on the mean score, risk perception was categorized as high if scored above the mean score and low if scored at the mean or less.

The reliability (internal consistency) of the questionnaire was checked based on the results of the pretest. The Cronbach's alpha was 0.703 for practice, 0.682 for knowledge, 0.679 for risk perception, and 0.764 for self-efficacy items.

Data were collected by health care professionals who had a bachelor's degree qualification, and prior data collection experiences. The data collectors and supervisors were trained on the data collection tool, the objective of the study, how to ensure confidentiality of information, and how to prevent transmission of COVID-19 during the interview. To reduce the risk of COVID-19 transmission during data collection, the data collectors used necessary personal protective equipment (PPE).

## Data processing and analysis

The collected data were manually checked for completeness, entered into Epi data manager version 4.0.2 and exported to SPSS version 22 for analysis. Descriptive statistics were done for different variables and bivariate binary logistic regression analysis was done to select candidate variables for multivariable binary logistic regression analysis at p value < 0.25 [26]. Finally, multivariable logistic regression analysis was done to control for the effect of possible confounders, and variables with p value < 0.05 were taken as statistically significant determinants of COVID-19 preventive behavior. Model fitness was evaluated using the Hosmer-Lemeshow goodness of fit test, and multicollinearity was checked using variance inflation factor (VIF).

## Results

### Socio-demographic characteristics

From the 422 total sample size, 416 participated in the study, resulting in a 98.6% response rate. The mean age of respondents was 27.26 (SD = ±8.35) years and more than half were aged 18–25 years. More than half (54.1%) of the study participants were female. The majority (84.1%) of the participants were single, and 44% of the participants had attended primary

**Table 1. Socio-demographic characteristics of waiters working in food and drinking establishments, Southwest Ethiopia, 2020.**

| Variables | Categories | Frequency | Percent |
|---|---|---|---|
| Age group | 18–25 | 218 | 52.4 |
| | 26–35 | 142 | 34.1 |
| | >35 | 56 | 13.5 |
| Sex | Male | 191 | 45.9 |
| | Female | 225 | 54.1 |
| Marital status | Single | 350 | 84.1 |
| | Married | 24 | 5.8 |
| | Divorced/ Widowed | 42 | 10.1 |
| Religion | Orthodox | 283 | 68.0 |
| | Muslim | 61 | 14.7 |
| | Protestant | 72 | 17.3 |
| Educational status | No education | 79 | 19.0 |
| | Primary | 183 | 44.0 |
| | Secondary/ Above | 154 | 37.0 |
| Ethnicity | Kafa | 165 | 39.7 |
| | Amhara | 110 | 26.4 |
| | Gurage | 36 | 8.7 |
| | Bench | 34 | 8.2 |
| | Tigre | 18 | 4.3 |
| | Oromo | 18 | 4.3 |
| | Sheka | 10 | 2.4 |
| | Meinit | 9 | 2.2 |
| | Others* | 9 | 2.2 |
| How many people live in your house? | Live alone | 86 | 20.7 |
| | Live with one or more persons | 330 | 79.3 |

Others*: Silte, Sheko and Dawuro.

school. Three hundred thirty (79.3%) study participants were living with one or more persons (Table 1).

## Knowledge of COVID-19

All study participants heard about the COVID-19 pandemic. The main sources of information were television (99.5%), followed by short messages from telecommunication (60.8%) and radio (58.9%) (Fig 1). The majority (84.4%) of the respondents know the cause of the new coronavirus disease. Three hundred twenty-eight (78.8%) study participants mentioned direct contact with infected people as a mode of transmission for the virus. Nearly three-fourths (72.8%) thought that contact with contaminated animals and 60.6% thought that touching contaminated objects/surfaces represent modes of spread for COVID-19. Furthermore, cough (86.6%), fever (83.2%), and shortness of breath (48.1%) were the commonly known symptoms of COVID 19. Regarding preventive methods, about 82.7% know frequent handwashing with soap and water or using alcohol-based sanitizer helps prevent COVID-19. Also, 75.5% of the respondents know that keeping social distance prevents COVID-19 (Table 2).

The mean cumulative score of knowledge was 10.32 (SD = 2.37), and the minimum and maximum scores were 3 and 15 respectively. Regarding compressive knowledge of COVID, 30.8% of the respondents had poor knowledge and 27.6% had good knowledge (Fig 2).

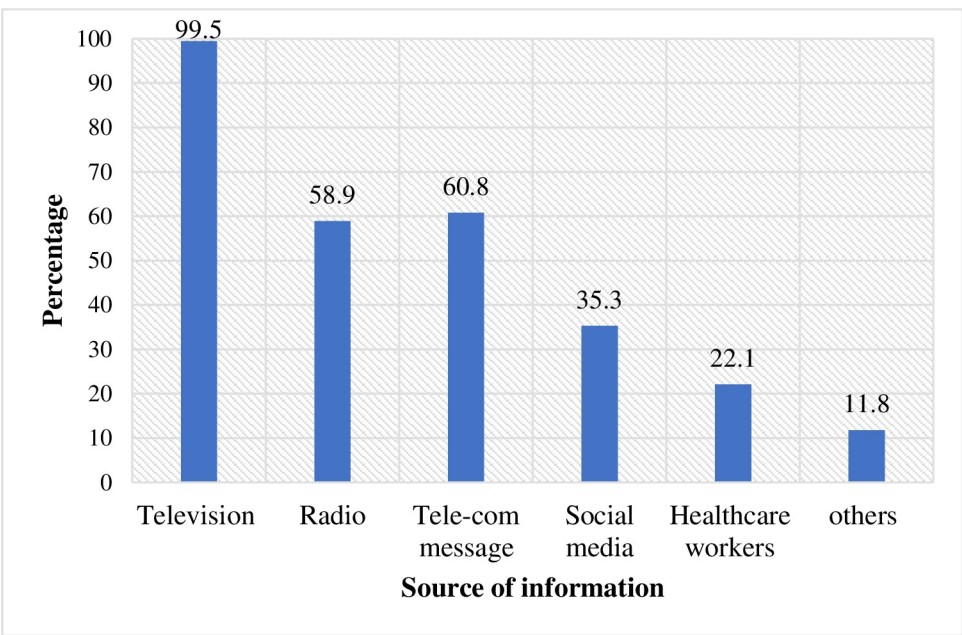

**Fig 1. Source of information about COVID-19 among waiters working in food and drinking establishments in Southwest Ethiopia, 2020 (n = 416).**

**Table 2. Knowledge of COVID-19 among waiters working in food and drinking establishments, Southwest Ethiopia, 2020.**

| Variables (n = 416) | Correct | Incorrect |
|---|---|---|
| | N- (%) | N- (%) |
| **What causes a new coronavirus disease?** | 351(84.4) | 65(15.6) |
| **Can COVID-19 virus transmit through the following routes/modes?** | | |
| Droplets from infected people | 250(60.1) | 166(39.9) |
| airborne | 206 (49.5) | 210(50.5) |
| Direct contact with an infected person | 328 (78.8) | 88 (21.2) |
| Touching of contaminated objects/surfaces | 252 (60.6) | 164 (39.4) |
| Contact with contaminated animals | 303 (72.8) | 113(17.2) |
| Mosquito bites | 395(95.0) | 21(5.0) |
| **Can COVID-19 infected patients present with the following symptoms?** | | |
| Fever | 346 (83.2) | 70 (16.8) |
| Cough | 361(86.8) | 55 (13.2) |
| Shortness of breath | 200(48.1) | 216 (51.9) |
| **Can COVID-19 virus be prevented by the following methods?** | | |
| Frequent hand washing using soap and water or alcohol-based sanitizer | 344 (82.7) | 72 (17.3) |
| Avoid close contact with anyone who has a fever and cough | 314 (75.5) | 102 (24.5) |
| Avoid unprotected direct contact with live animals and surfaces | 156(62.5) | 260 (37.5) |
| Sleeping under the mosquito net | 397 (95.4) | 19 (4.6) |
| **Is there effective treatment or vaccine for the COVID-19 currently?** | 312 (75.0) | 104(25.0) |

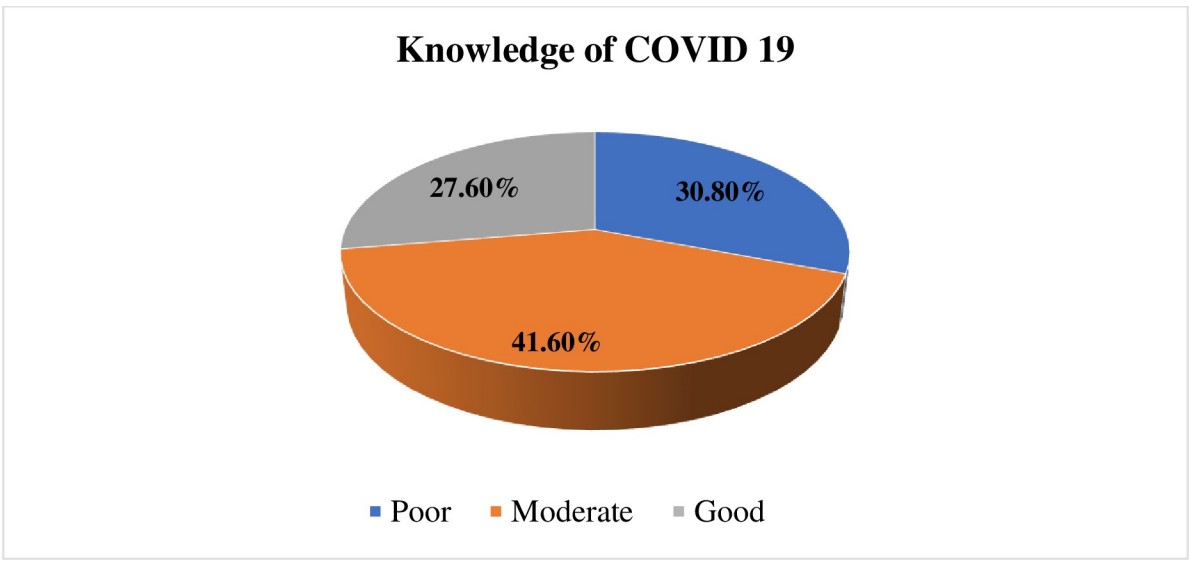

**Fig 2. Comprehensive knowledge about COVID 19 among waiters working in food and drinking establishments in Southwest Ethiopia, 2020 (n = 416).**

## Preventive behavior of COVID 19

About 27% of the respondents never keep physical distance, while only 5% keep physical distance always. About 18.8% and 19.2% of the respondents wash their hands with water and soap or with sanitizer frequently and always, respectively. Only 6% of the respondents using facemasks at work or outside their home always, and about 6.7% and 4% were using gloves frequently and always, respectively (Table 3). The overall proportion of study participants with good COVID-19 preventive behavior were only 21.2%. The common barriers reported were the difficulty of maintaining social distance at work due to the nature of their work, shortage of facemasks and gloves, and high efforts needed to implement prevention measures (Fig 3).

## Risk perception and self-efficacy to practice preventive measures

Two hundred twenty-two (53.4%) of the participants had a high-risk perception towards COVID-19, while the remaining 46.6% had a low-risk perception. Besides, 53.1% of study participants had high self-efficacy to practice COVID-19 preventive methods.

**Table 3. Preventive behavior of COVID-19 virus among waiters working in food and drinking establishments, Southwest Ethiopia, 2020.**

| Questions | Never | Rarely | Sometimes | Frequently | Always |
|---|---|---|---|---|---|
| | N (%) | N (%) | N (%) | N (%) | N (%) |
| How often are you maintaining physical distance? | 22(5.3) | 39(9.4) | 112(26.9) | 151(36.3) | 92(22.1) |
| How often are you avoiding larger gatherings? | 57(13.7) | 58(13.9) | 94(22.6) | 110(26.4) | 97(23.3) |
| How often are you avoiding touching your face, eyes, mouth, and nose? | 73(17.5) | 78(18.8) | 122(29.3) | 68(16.3) | 75(18) |
| How often are you washing your hands with water and soap or sanitizers? | 111(26.7) | 45(10.8) | 78(18.8) | 102(24.5) | 80(19.2) |
| How often are you avoiding contact with people who had fever and cough? | 110(26.4) | 38(9.1) | 60(14.4) | 110(26.4) | 98(23.6) |
| How often are you wearing a facemask when you are at work or outside the home | 161(38.7) | 74(17.8) | 108(26.0) | 48(11.5) | 25(6.0) |
| How often do you use public transportation during the months of the pandemic? | 118(28.4) | 67(16.1) | 124(29.8) | 46(11.1) | 61(14.7) |
| How often you avoid unprotected contacting (touching) of frequently contacted surfaces | 68(16.3) | 65(15.6) | 126(30.3) | 99(23.8) | 58(13.9) |
| Did you avoid unnecessary travel during the months of the pandemic? | 178(42.8) | 57(13.7) | 74(17.8) | 42(10.1) | 65(15.6) |
| How often are you wearing a glove at work? | 189(45.4) | 72(17.3) | 110(26.4) | 28(6.7) | 17(4.1) |

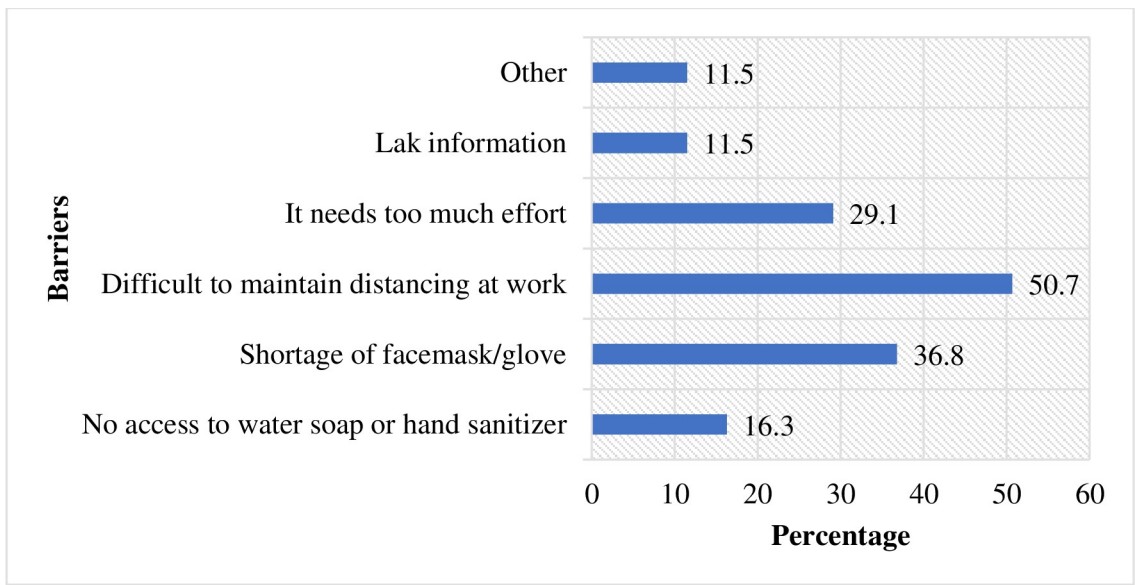

**Fig 3. Barriers or hindering factors to implement COVID-19 prevention measures among waiters working in food and drinking establishments in Southwest Ethiopia, 2020 (n = 416).**

### Factors associated with COVID-19 preventive behavior

In simple binary logistic regression analysis, sex, marital status, educational status, number of people living in the house, risk perception, and perceived self-efficacy had a p value $\leq 0.25$; hence, they were candidates for the multivariable binary logistic model. However, age and knowledge about COVID-19 were excluded from the multivariable model because their p-values were greater than 0.25 in the bivariate binary logistic regression analysis. The number of people living in the home was excluded from the model because the model best fit when this variable removed from the model. In the final multivariable binary logistic regression model, sex, educational status, risk perception, and perceived self-efficacy were significantly associated with COVID-19 preventive behavior (p <0.05) (Table 4).

The odds of good COVID-19 preventive behavior among females was 2.33 times higher than their counterparts (AOR = 2.33, 95% CI: 1.38–3.94). The odds of good COVID-19 preventive behavior among waiters who had no formal education was about 0.39 times lower compared to those who attended at least secondary education (AOR = 0.39, 95% CI: 0.17–0.88). The odds of good COVID-19 preventive behavior among waiters who had high-risk perception toward COVID-19 was 2.56 times higher compared to those who had low-risk perception toward COVID-19 (AOR = 2.26, 95% CI: 1.51–4.32). The odds of good COVID-19 preventive behavior among waiters who had high perceived self-efficacy was almost twice as high as those who had low self-efficacy to practice COVID-19 prevention methods (AOR = 1.1.75, 95% CI: 1.05–2.90).

### Discussion

After the first case of COVID-19 was detected in Ethiopia on March 13[th], 2020 [3], the government started responding to the pandemic aggressively. A state of emergency was declared to counter and control the spread of COVID-19 and mitigate its impact. During the periods of the state of emergency: land borders were closed, schools and universities remained shut, all

**Table 4.  Factors associated with COVID 19 preventive behavior among waiters working in food and drinking establishments, Southwest Ethiopia, 2020.**

| Variables | Preventive behavior | | COR (95% CI) | AOR (95% CI) | P-value |
|---|---|---|---|---|---|
| | Good (%) | Poor (%) | | | |
| Age group | | | | | |
| 18–25 | 48(21.1) | 180(78.9) | 1 | 1 | |
| 26–35 | 26(19.0) | 111(81.0) | 0.88(0.52–1.50) | 0.81(0.45,1.43) | 0.462 |
| >35 | 14(27.5) | 37(72.5) | 1.42(0.71–2.84) | 1.49(0.72,3.07) | 0.284 |
| Sex | | | | | |
| Male | 27(14.1) | 164(85.9) | 1 | 1 | |
| Female | 61(27.1) | 167(72.9) | 2.26(1.37–3.73) | 2.33(1.38–3.94) | 0.002 |
| Marital status | | | | | |
| Single | 74(21.1) | 276(78.9) | 1 | 1 | |
| Married | 8(33.3) | 16(66.7) | 1.86(0.76–4.53) | 2.26(0.89,5.73) | 0.087 |
| Divorced/ Widowed | 6(14.3) | 36(85.7) | 0.62(0.25–1.53) | 0.69(0.27,1.77) | 0.444 |
| Educational status | | | | | |
| No education | 9(11.4) | 70(88.6) | 0.48(0.17–0.83) | 0.39(0.17–0.88) | 0.023 |
| Primary | 40(21.9) | 143(78.1) | 0.82(0.49–1.37) | 0.99(0.58–1.69) | 0.978 |
| Secondary/ Above | 39(25.3) | 115(74.7) | 1 | 1 | |
| Knowledge of COVID 19 | | | | | |
| Poor | 25(19.5) | 103(80.5) | 1 | 1 | |
| Fair | 36(20.8) | 137(79.2) | 0.79(0.42–1.46) | 0.96(0.49,1.89) | 0.911 |
| Good | 27(23.5) | 88(76.5) | 0.86(0.48–1.51) | 0.98(0.53,1.82) | 0.949 |
| Risk perception | | | | | |
| Low | 27(13.9) | 167(86.1) | 1 | 1 | |
| High | 61(27.5) | 161(72.5) | 2.34(1.42–3.87) | 2.56(1.51–4.32) | <0.001 |
| Self-efficacy | | | | | |
| Low | 38(17.2) | 183(82.8) | 1 | 1 | |
| High | 50(25.6) | 145(74.4) | 1.66(1.03–2.67) | 1.75(1.05–2.90) | 0.030 |

Hosmer and Lemeshow test; $X2 = 9.36$; df = 8; p = 0.313.

gathering of more than four persons were forbidden unless specially permitted, all vehicles (public and private) and railway, and light railway allowed to operate only at 50% and 25% of passenger capacity, respectively. Moreover, hotels, restaurants, and cafeterias didn't allow service to more than three patrons at a single table and should ensure that tables that are being used by patrons simultaneously are at least two adult strides apart [27]. However, still date a full lockdown has not been declared in Ethiopia. Hotels, bars, restaurants, and cafeterias are not completely closed during the pandemic in Ethiopia; hence, the risk of the spread of the virus is high because they are visited by many people who interact among themselves and with employees. Therefore, every staff must strictly follow the basic protective measures.

This study aimed to assess knowledge of COVID-19 and the practice of preventive behaviors among waiters in southwest Ethiopia. The finding of this study showed that the majority of the respondents knew the cause of COVID-19. It was revealed that a significant proportion of participants know the mode of spread of the virus (inhalation of droplets from infected people, direct contact with infected people, contaminated animals, and contaminated objects/surfaces). A few participants thought that COVID-19 can be transmitted by mosquito bites. A study conducted in Nigeria also reported a similar misconception [28]. Furthermore, the majority of the respondents know common symptoms of COVID-19 disease and its prevention methods. The main source of information for study participants was from television.

It was also identified that only 21.2% of the study participants had good COVID-19 preventive behavior. This finding is almost similar to a study conducted in Myanmar [29]. However, this finding is lower than a study conducted in Northwest Ethiopia, and Pakistan [30–32]. This variation could be due to the differences in socio-economic characteristics of study participants or it might be due to disparity in access to media. The current study was conducted in remote areas of the country where access to the internet and electronic media is relatively low. Also, it might be due to barriers related to the nature of the work, shortage of facemasks and gloves during early phase of the outbreak. The low COVID-19 preventive behavior among waiters has a great public health implication; if waiters are infected with COVID-19, there will be a high risk of spreading the disease to the community because they frequently come in contact with many people due to the nature of their work.

The odds of good practice of COVID-19 prevention methods among females was higher compared to males. Most of the time females are more careful of themselves and others around them. Other studies also indicated that males are less likely to practice COVID-19 prevention methods compared to females [33, 34]. Thus, behavioral intervention programs had better consider women as change agents in adopting a particular preventive behavior in the family and the community as well.

This study identified that a higher score of COVID-19 preventive behavior is associated with higher risk perception. This finding is consistent with findings from other studies [4, 14]. This could be due to individuals who perceive they are at high risk might engage in preventive behaviors. This may imply it could be possible to enhance the desired behavior by proper risk communication about the disease.

The educational status of participants was also associated with COVID-19 preventive behavior. The odds of good COVID-19 preventive behavior among study participants who had no formal education was lower compared to those who attended at least secondary education. Similarly, a study conducted among visitors in Jimma University Medical Center in Ethiopia showed that proper handwashing with soap and water or sanitizer was negatively associated with lower educational status [35]. This could be because educated people are in a better position to have access to COVID-19 related information. Moreover, they could easily comprehend instruction/recommendations made by health personnel, media, and other relevant bodies.

The odds of good COVID-19 preventive behavior among subjects who had high perceived self-efficacy was higher when compared to those with low self-efficacy to practice COVID-19 prevention methods. A similar claim was forwarded by a previous study conducted elsewhere [33]. This is substantiated by the theory of self-efficacy which states that self-efficacy influences every human endeavor; by determining the belief a person holds regarding his or her ability to perform a particular action [36]. Thus, enabling people to enhance their self-efficacy through various individualized approach could bring the desired behavioral changes.

## Limitation of the study

Due to the cross-sectional nature of this study, it is difficult to establish cause-effect relationships. Also, since data were collected through self-reports, there might be a risk of desirability bias. Furthermore, due to the fact that the study design is purely quantitative, we cannot explain the reasons for the observed effect and their meanings in that particular context.

## Conclusion

A significant proportion of waiters knew common symptoms of COVID 19, route of transmission, and its prevention methods. However, good COVID-19 preventive behavior was very

low. It was recognized that being a female, higher schooling, having a high-risk perception, and having a high perceived self-efficacy were positively associated with good COVID-19 preventive behavior. Thus, all concerned bodies working on the prevention and control of COVID-19 should give attention to this population to enhance compliance with recommended preventive behaviors through addressing these significant predictors. Since widespread infection among waiters with COVID-19 has an important public health implication, law enforcement bodies shall employ a compulsory mechanism to monitor the implementation of all recommendations.

## Supporting information

**S1 Dataset.**
(SAV)

**S1 File.**
(DOCX)

## Acknowledgments

We would like to express our heartfelt gratitude to Mizan-Tepi University for providing us an ethical clearance to undertake this research project. Also, we thanks the study participants and data collectors for their valuable contributions.

## Author Contributions

**Conceptualization:** Qaro Qanche, Adane Asefa, Tadesse Nigussie, Shewangizaw Hailemariam, Tadesse Duguma.

**Data curation:** Qaro Qanche, Adane Asefa, Tadesse Nigussie, Shewangizaw Hailemariam, Tadesse Duguma.

**Formal analysis:** Qaro Qanche, Adane Asefa, Tadesse Nigussie, Tadesse Duguma.

**Funding acquisition:** Tadesse Duguma.

**Investigation:** Adane Asefa, Tadesse Nigussie, Shewangizaw Hailemariam, Tadesse Duguma.

**Methodology:** Qaro Qanche, Adane Asefa, Tadesse Nigussie, Shewangizaw Hailemariam, Tadesse Duguma.

**Project administration:** Qaro Qanche, Adane Asefa, Shewangizaw Hailemariam, Tadesse Duguma.

**Resources:** Qaro Qanche, Adane Asefa, Tadesse Nigussie, Shewangizaw Hailemariam, Tadesse Duguma.

**Software:** Qaro Qanche, Adane Asefa, Shewangizaw Hailemariam, Tadesse Duguma.

**Supervision:** Adane Asefa, Tadesse Nigussie.

**Validation:** Adane Asefa.

**Visualization:** Qaro Qanche, Adane Asefa, Tadesse Nigussie, Tadesse Duguma.

**Writing – original draft:** Adane Asefa, Shewangizaw Hailemariam.

**Writing – review & editing:** Qaro Qanche, Adane Asefa, Tadesse Nigussie, Shewangizaw Hailemariam, Tadesse Duguma.

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
