## [Decision Letter · Decision Letter 0]

28 Oct 2020

PONE-D-20-25994

Knowledge of COVID-19 and practice of preventive behaviors among waiters working in
food and drinking establishments in Southwest, Ethiopia

PLOS ONE

Dear Dr. Asefa,

Thank you for submitting your manuscript to PLOS ONE. After careful consideration, we
feel that it has merit but does not fully meet PLOS ONE’s publication criteria as it
currently stands. Therefore, we invite you to submit a revised version of the
manuscript that addresses the points raised during the review process.

Some revision of grammar is needed, and the discussion should be in context of
current measures that are in place in Ethiopia to prevent the transmission of
COVID-19. 

Please submit your revised manuscript by Dec 12 2020 11:59PM. If you will need more
time than this to complete your revisions, please reply to this message or contact
the journal office at plosone@plos.org. When
you're ready to submit your revision, log on to https://www.editorialmanager.com/pone/ and select the 'Submissions
Needing Revision' folder to locate your manuscript file.

If you would like to make changes to your financial disclosure, please include your
updated statement in your cover letter. Guidelines for resubmitting your figure
files are available below the reviewer comments at the end of this letter.

We look forward to receiving your revised manuscript.

Kind regards,

Jennifer A Hirst, DPhil

Academic Editor

PLOS ONE

Additional Editor Comments:

Please find some additional comments on the manuscript that need revising as
follows:

• please provide a copy of the questionnaire used

• More information on the questionnaire answers and scoring needs to be provided. For
example, were correct answers from multiple choice or was a judgement made by the
interviewer on whether the answer given was right or wrong?

English - needs proofreading

Introduction

Page 4 – “between guests, staff, and guests and staff”: “between guests and staff” is
sufficient.

Study population – first sentence needs revising

Page 6 sample size: please explain what the lottery method is

Page 6 – Measurements. How was the 60% threshold reached for good behaviour?

Similarly, how were the knowledge thresholds established?

Journal Requirements:

3. Thank you for stating in the text of your manuscript "Before data collection,
ethical clearance and a formal letter were obtained from the College of Health
Science and Mizan-Tepi University. Written informed consent was obtained from all
participants, after explaining the study’s purpose, risks, and benefits. Moreover,
participants were assured that their participation was voluntary and personal
information will not be disclosed to third parties." Please also add this
information to your ethics statement in the online submission form.

4. Please include additional information regarding the survey or questionnaire used
in the study and ensure that you have provided sufficient details that others could
replicate the analyses. For instance, if you developed a questionnaire as part of
this study and it is not under a copyright more restrictive than CC-BY, please
include a copy, in both the original language and English, as Supporting
Information.

"The authors would like to thank Mizan-Tepi University for supporting the
project."

"The author(s) received no specific funding for this work"

6. We note that you have indicated that data from this study are available upon
request. PLOS only allows data to be available upon request if there are legal or
ethical restrictions on sharing data publicly. For information on unacceptable data
access restrictions, please see http://journals.plos.org/plosone/s/data-availability#loc-unacceptable-data-access-restrictions.

7. Your ethics statement should only appear in the Methods section of your
manuscript. If your ethics statement is written in any section besides the Methods,
please move it to the Methods section and delete it from any other section. Please
ensure that your ethics statement is included in your manuscript, as the ethics
statement entered into the online submission form will not be published alongside
your manuscript.

Reviewers' comments:

Reviewer's Responses to Questions

**Comments to the Author**

1. Is the manuscript technically sound, and do the data support the conclusions?

Reviewer #1: Yes

Reviewer #2: Yes

2. Has the statistical analysis been performed
appropriately and rigorously? 

Reviewer #1: Yes

Reviewer #2: Yes

3. Have the authors made all data underlying the
findings in their manuscript fully available?

Reviewer #1: Yes

Reviewer #2: Yes

4. Is the manuscript presented in an intelligible
fashion and written in standard English?

Reviewer #1: No

Reviewer #2: Yes

5. Review Comments to the Author

Reviewer #1: Thank you for sharing this interesting research with me. Many research
papers focus on the knowledge and attitude of the general public or medical staff
towards COVID-19. This manuscript is focusing on waiters, who are also at risk of
getting or transmitting the virus. Thus, they should follow strict measures to
protect themselves as well as others.

General comments:

The language is clear in most of the paragraphs. However, some sentences are not
clear and contain grammar mistakes. So, the manuscript may benefit from language
proofreading.

More depth should be added to some parts of the discussion.

Specific comments

Introduction

• Paragraph 1, line 2: Please add the full name of the virus (SARS-CoV-2)

• Paragraph 1, line 2: You mentioned that " It first occurred in the Hubei province
of China in December 2019" Please modify into " It was first discovered in the Hubei
province of China in December 2019"

• Paragraph 2, line 3: You mentioned that "The disease clinically presented with

Fever, cough, difficulty breathing, and other flu-like signs and symptoms including
runny and stuffy nose, sneezing, and sore throat." The disease may present in sever
and fatal forms as well, and that's why it's an international concern.

Methods

• In general, the methods are detailed and well written.

Results

• In the first paragraph about knowledge about COVID-19, you mentioned that "Nearly
three-fourths (72.8%) were mentioned contact with contaminated animals and 60.6%
mentioned directly touching contaminated objects/surfaces as modes spread of COVID
19." I would prefer if phrase the sentence as follows " Nearly three-fourths (72.8%)
thought that contact with contaminated animals and 60.6% thought that touching
contaminated objects/surfaces represent modes spread of COVID 19."

• Table 2: First variable: Cause new corona disease should be changed into "The
disease is caused by a new corona virus."

• The table should divided into titles (in bald) and subtitles. For example Knowledge
of mode of transmission should be bald followed by the questions about this point.
The same for symptoms.

• Table 3: The first option in the heading of the table should be "never" not
"none"

• Table 3: Question 4 : Please modify into "How often are you washing your

hands with water and soap or sanitizers?

• Table 3: Questions 7 and 9 don't make sense to me. Instead you could have asked
"How often do you use public transportation?" and "Did you avoid unnecessary travel
during the months of the pandemic?"

• Table 4: Please add the p value for each factor of the regression analysis to the
table.

Discussion

• I would prefer if the discussion starts with a brief paragraph about the measures
taken by the government in Ethiopia to prevent the transmission of COVID-19. For
example: Was there complete lockdown? Were restaurants and bars closed? If the
answer is yes, when were they opened again? What are the measures imposed by the
government on restaurants and bars to limit the spread of infection? Did these
restaurants applied strict measures on quests as well? This will be very useful.

• Paragraph 1, line 8: You mentioned that contaminated animals are a source of
infection. Please add a reference for this information.

• The study showed that majority of respondents knew common symptoms of COVID 19
disease and its prevention methods. It would have been interesting if they were
asked about their source of information, especially because some of them had
misconceptions about the sources of transmission.

• Paragraph 2 line 1: You mentioned that "It was also identified that only 21.2% of
the study participants had good preventive behavior against COVID-19." Was this due
to limited tools? e.g no masks…no gloves...no soap? Please elaborate.

• Paragraph 2 line 4: mentioned that "This variation could be due to differences in
the socio-economic characteristics of study participants or it might be due to
disparity in media exposure as the study area is relatively far from the center."
What type of media channels? And why there is disparity in media exposure? No
internet? No electricity?

• Paragraph 4: You mentioned that "The study identified that a higher score of
COVID-19 preventive practice associated with higher risk perception." Why do you
think this group had a higher risk perception? Maybe they had a family member who
has been infected or had chronic diseases?

• The same applies for paragraph 6, where You mentioned that " The odds of good
practice of COVID-19 prevention methods among subjects who had high perceived
self-efficacy was higher as compared to those with low self-efficacy to practice
COVID-19 prevention methods." Why did this group had higher self efficacy?

• Was there a difference between application of preventive practices between those
working in hotels and bars for example? Did different levels of restaurants (fantasy
versus popular restaurants) affected using protective measures?

Reviewer #2: in Data processing and analysis paragraph , please write manufacture
origin of SPSS,

please add references for you paragraph that you describe multivariate logistic
regression analysis model , ( Bivariate binary logistic regression analysis was
done

to select candidate variables for multivariable binary logistic regression analysis
at p-value < 0.25)

6. PLOS authors have the option to publish the peer
review history of their article (what does this mean?). If published, this will
include your full peer review and any attached files.

If you choose “no”, your identity will remain anonymous but your review may still be
made public.

**Do you want your identity to be public for this peer review?** For
information about this choice, including consent withdrawal, please see our
Privacy Policy.

Reviewer #1: **Yes: **Ahmed Samir Abdelhafiz

Reviewer #2: **Yes: **wafaa Yousif Abdel Wahed

---

## [Author Response · Author response to Decision Letter 0]

23 Nov 2020

Rebuttal letter

Dear Editor,

Dear Reviewers,

Thank you very much for your important observations and recommendations, which, in
our opinion, are contributing to a significant improvement of our manuscript’s
quality and scientific impact. Based on your recommendations, we addressed all the
issues raised and the corrections are included in the revised version of the
manuscript.

In the following, we provide details of the changes added to the manuscript, in
respect to your valuable comments.

Editor Comments:

Please find some additional comments on the manuscript that need revising as
follows:

1. Please provide a copy of the questionnaire used

Response: Thank for the request. We have uploaded questionnaire as “Supporting
Information files”.

2. More information on the questionnaire answers and scoring needs to be provided.
For example, were correct answers from multiple choice or was a judgement made by
the interviewer on whether the answer given was right or wrong?

Response: Thank you. We included detail about questionnaire under method section;
measures sub-headings (page 6). The questionnaire that used to measure the knowledge
are based yes, no or I don’t know responses. The correct answers were determined in
accordance with the available evidence during early phase of the pandemic. 

3. English - needs proofreading

Response: Thank you for your observation. We have edited the whole manuscript
thoroughly for language error. 

4. Page 4 – “between guests, staff, and guests and staff”: “between guests and staff”
is sufficient

Response. Thank you for your suggestion. We have accepted it 

5. Study population – first sentence needs revising

Response. Thank you for your observation. We have revised the statement (page 5). 

6. Page 6 sample size: please explain what the lottery method is

Response. Thanks. We have now described how lottery method done on page 5 and 6.

7. Page 6 – Measurements. How was the 60% threshold reached for good behaviors?

Response. Thank you, the question. A 60% cutoff point for good behavior was on the
modified threshold from study done in Iran. We have also considered modified Bloom’s
cutoff point (60% cutoff point)

8. Similarly, how were the knowledge thresholds established?

Response. Thank you, again. The threshold was determined according original Bloom’s
cutoff point. We have cited the reference for the threshold used. 

Journal Requirements:

Response: Now we have updated the whole manuscript as per the requirement by the
journal. 

Response: Thank you for your observation. We have edited the whole manuscript
thoroughly for language error. 

3. Please add ethics statement in the online submission form.

Response: thanks, we have done it. 

4. Please include additional information regarding the survey or questionnaire used
in the study and ensure that you have provided sufficient details that others could
replicate the analyses. For instance, if you developed a questionnaire as part of
this study and it is not under a copyright more restrictive than CC-BY, please
include a copy, in both the original language and English, as Supporting
Information.

Response: We included more detail about survey tool in method section under measures
sub-headings (page 6). We have also uploaded the survey to which prepared as
Supporting Information

Thank you for stating the following in the Acknowledgments Section of your
manuscript:

5. "The authors would like to thank Mizan-Tepi University for supporting the
project."

Response: Sorry for unclarity in our statement. We did not obtain any formal funding
form any institution or body. The research was done as part of our professional
duties in our institution. We acknowledged Mizan-Tepi university for providing us
ethical clearance and informal supports. We have no modified the statements in the
acknowledgement section. 

6. We note that you have indicated that data from this study are available upon
request. PLOS only allows data to be available upon request if there are legal or
ethical restrictions on sharing data publicly. For information on unacceptable data
access restrictions, please see http://journals.plos.org/plosone/s/data-availability#loc-unacceptable-data-access-restrictions. 

a. If there are ethical or legal restrictions on sharing a de-identified data set,
please explain them in detail (e.g., data contain potentially identifying or
sensitive patient information) and who has imposed them (e.g., an ethics committee).
Please also provide contact information for a data access committee, ethics
committee, or other institutional body to which data requests may be sent.

b. If there are no restrictions, please upload the minimal anonymized data set
necessary to replicate your study findings as either Supporting Information files or
to a stable, public repository and provide us with the relevant URLs, DOIs, or
accession numbers. Please see http://www.bmj.com/content/340/bmj.c181.long for guidelines on how
to de-identify and prepare clinical data for publication. For a list of acceptable
repositories, please see http://journals.plos.org/plosone/s/data-availability#loc-recommended-repositories.

Response: There is no ethical or legal restriction on sharing of anonymized data set.
Thus, we have uploaded data set Supporting Information files. 

7. Your ethics statement must appear in the Methods section of your manuscript. If
your ethics statement is written in any section besides the Methods, please move it
to the Methods section and delete it from any other section. Please also ensure that
your ethics statement is included in your manuscript, as the ethics section of your
online submission will not be published alongside your manuscript. 

Response: Thanks for your observation. Now we include the ethics statement only in
methods section. 

Reviewer #1

1. General comments:

1.1. The language is clear in most of the paragraphs. However, some sentences are not
clear and contain grammar mistakes. So, the manuscript may benefit from language
proofreading.

Response: Thank you for your observation. We have edited the whole manuscript
thoroughly for language error. 

2. Specific comments

Introduction

2.1. Paragraph 1, line 2: Please add the full name of the virus (SARS-CoV-2). 

Response: Thank you for your comment. We have written full name of SARS-CoV-2 in
introduction, first paragraph (page 3, line 5)

2.2. Paragraph 1, line 2: You mentioned that " It first occurred in the Hubei
province of China in December 2019" Please modify into " It was first discovered in
the Hubei province of China in December 2019"

Response: Thanks. We accepted the suggestion. (page 3, line 5) 

2.3. Paragraph 2, line 3: You mentioned that "The disease clinically presented with.
Fever, cough, difficulty breathing, and other flu-like signs and symptoms including
runny and stuffy nose, sneezing, and sore throat." The disease may present in sever
and fatal forms as well, and that's why it's an international concern.

Response: The statements are modified as follow; “The disease is clinically presented
with fever, cough, difficulty breathing, and other flu-like signs and symptoms
including runny and stuffy nose, sneezing, and sore throat. Most COVID-19 patients
can recover with mild or no symptoms; however, in rare cases, patients may develop a
severe acute respiratory syndrome that requires mechanical ventilation”. Paragraph
2, line 15-19 (page 3) 

Results

2.4. In the first paragraph about knowledge about COVID-19, you mentioned that
"Nearly three-fourths (72.8%) were mentioned contact with contaminated animals and
60.6% mentioned directly touching contaminated objects/surfaces as modes spread of
COVID 19." I would prefer if phrase the sentence as follows " Nearly three-fourths
(72.8%) thought that contact with contaminated animals and 60.6% thought that
touching contaminated objects/surfaces represent modes spread of COVID 19."

Response: Thank you for your suggestion. We accepted the suggestion and included in
manuscript (page 9; line 7-9) 

2.5. Table 2: First variable: Cause new corona disease should be changed into "The
disease is caused by a new corona virus."

Response: Thank you modified the statement to interrogative form; What causes a new
coronavirus disease?

2.6. The table should divide into titles (in bald) and subtitles. For example,
Knowledge of mode of transmission should be bald followed by the questions about
this point. The same for symptoms

Responses: Thank for your suggestions. To avoid redundance of words and minimizing
the crowding of table with we to put common questions as in bold. For example, “Can
COVID-19 infected patients present with the following symptoms?”

2.7. Table 3: The first option in the heading of the table should be "never" not
"none"

Responses: We accepted the suggestion. 

2.8. Table 3: Question 4: Please modify into "How often are you washing your hands
with water and soap or sanitizers? Table 3: Questions 7 and 9 don't make sense to
me. Instead you could have asked "How often do you use public transportation?" and
"Did you avoid unnecessary travel during the months of the pandemic

Response: Thank you for constructive suggestions. We accepted and incorporated all in
table 3. 

2.9. Table 4: Please add the p value for each factor of the regression analysis to
the table.

Response: We accepted the comment and add p-value for variables in final best fit
model. 

 Discussion

1. I would prefer if the discussion starts with a brief paragraph about the measures
taken by the government in Ethiopia to prevent the transmission of COVID-19. For
example: Was there complete lockdown? Were restaurants and bars closed? If the
answer is yes, when were they opened again? What are the measures imposed by the
government on restaurants and bars to limit the spread of infection? Did these
restaurants applied strict measures on quests as well? This will be very useful.

Response: Thank so much for the comments. We have addressed the comments first
paragraph of the discussion. 

2. Paragraph 1, line 8: You mentioned that contaminated animals are a source of
infection. Please add a reference for this information.

Response: We thanks for the suggestion. Now, the issue is described with reference in
introduction part page 3, line 11-15

3. The study showed that majority of respondents knew common symptoms of COVID 19
disease and its prevention methods. It would have been interesting if they were
asked about their source of information, especially because some of them had
misconceptions about the sources of transmission.

Response: Thank for your observation. Since we have data regarding source of
information, we have now included it result part page 9, line 5-7

4. Paragraph 2 line 1: You mentioned that "It was also identified that only 21.2% of
the study participants had good preventive behavior against COVID-19." Was this due
to limited tools? e.g no masks…no gloves...no soap? Please elaborate.

Response: Thank for your nice observation. Results regarding barrier/hindering
factors are added in both results and discussions sections 

5. Paragraph 2 line 4: mentioned that "This variation could be due to differences in
the socio-economic characteristics of study participants or it might be due to
disparity in media exposure as the study area is relatively far from the center."
What type of media channels? And why there is disparity in media exposure? No
internet? No electricity?

Response: Thank you. The study was conducted in remote area of the country where
health and infrastructure in very low. Thus, access of health information health
information is relatively low the area due low internet coverage and electronic
medias. 

6. Paragraph 4: You mentioned that "The study identified that a higher score of
COVID-19 preventive practice associated with higher risk perception." Why do you
think this group had a higher risk perception? Maybe they had a family member who
has been infected or had chronic diseases?

Response: since the study design purely quantitative, we could fully explain the
reason behind high risk perception. Thus, we have described this in limitation of
the study. Page 17

7. The same applies for paragraph 6, where You mentioned that " The odds of good
practice of COVID-19 prevention methods among subjects who had high perceived
self-efficacy was higher as compared to those with low self-efficacy to practice
COVID-19 prevention methods." Why did this group had higher self-efficacy? 

Response: Due to the quantitative nature this study we did not know why those group
have high self-efficacy. 

8. Was there a difference between application of preventive practices between those
working in hotels and bars for example? Did different levels of restaurants (fantasy
versus popular restaurants) affected using protective measures?

Response: In the area, many establishments provide mixed service. For instance,
hotels are providing bar and restaurant service and vice-versa. Thus, it is
difficult to assess if there are variations in practice of protective measures. 

Reviewer #2

1. In Data processing and analysis paragraph, please write manufacture origin of
SPSS.

Response: thank you. We have accepted the suggestion and included in the
manuscript.

2. Please add references for you paragraph that you describe multivariate logistic
regression analysis model, ( Bivariate binary logistic regression analysis was done
to select candidate variables for multivariable binary logistic regression analysis
at p-value < 0.25)

Response: thank you. We have cited the reference as per your recommendation.

---

## [Decision Letter · Decision Letter 1]

16 Dec 2020

PONE-D-20-25994R1

Knowledge of COVID-19 and preventive behaviors among waiters working in food and
drinking establishments in Southwest Ethiopia

PLOS ONE

Dear Dr. Asefa,

Thank you for submitting your manuscript to PLOS ONE. After careful consideration, we
feel that it has merit but does not fully meet PLOS ONE’s publication criteria as it
currently stands. Therefore, we invite you to submit a revised version of the
manuscript that addresses the points raised during the review process.

Please submit your revised manuscript by Jan 30 2021 11:59PM. If you will need more
time than this to complete your revisions, please reply to this message or contact
the journal office at plosone@plos.org. When
you're ready to submit your revision, log on to https://www.editorialmanager.com/pone/ and select the 'Submissions
Needing Revision' folder to locate your manuscript file.

If you would like to make changes to your financial disclosure, please include your
updated statement in your cover letter. Guidelines for resubmitting your figure
files are available below the reviewer comments at the end of this letter.

We look forward to receiving your revised manuscript.

Kind regards,

Jennifer A Hirst, DPhil

Academic Editor

PLOS ONE

Reviewers' comments:

Reviewer's Responses to Questions

**Comments to the Author**

1. If the authors have adequately addressed your comments raised in a previous round
of review and you feel that this manuscript is now acceptable for publication, you
may indicate that here to bypass the “Comments to the Author” section, enter your
conflict of interest statement in the “Confidential to Editor” section, and submit
your "Accept" recommendation.

Reviewer #1: (No Response)

2. Is the manuscript technically sound, and do the data
support the conclusions?

Reviewer #1: Yes

3. Has the statistical analysis been performed
appropriately and rigorously? 

Reviewer #1: Yes

4. Have the authors made all data underlying the
findings in their manuscript fully available?

Reviewer #1: Yes

5. Is the manuscript presented in an intelligible
fashion and written in standard English?

Reviewer #1: Yes

6. Review Comments to the Author

Reviewer #1: Thank you for making the required changes. The manuscript has improved a
lot. I have some minor comments. Kindly find the comments below

• Page 3, line 19: You mentioned that "however, in rare cases, patients may develop a
severe acute respiratory syndrome that requires mechanical ventilation" I think that
"rare" is not the correct term here. Maybe you can add a reference which includes
the percentage of cases that develop ARDS.

• Table 3, question one: The first question should be modified into "How often are
you maintaining physical distance?

• Table 4: The p values for some parameters are not written. Please add them.

Discussion

• Page 14, line 11: You mentioned that " During the periods of the state of
emergency: land borders are closed, schools and universities remain shut, all
gathering of more than four persons are forbidden" Please use the past tense instead
of the present tense " borders were closed....universities remained shut… etc"
please do this all through the paragraph.

• Page 14, line 17: You mentioned that "Moreover, hotels, restaurants, and cafeterias
allowed services to more than three patrons at a single table" I think you mean that
that they didn't allow service to more than 3 persons at a single table. If so,
please correct the sentence.

7. PLOS authors have the option to publish the peer
review history of their article (what does this mean?). If published, this will
include your full peer review and any attached files.

If you choose “no”, your identity will remain anonymous but your review may still be
made public.

**Do you want your identity to be public for this peer review?** For
information about this choice, including consent withdrawal, please see our
Privacy Policy.

Reviewer #1: **Yes: **Ahmed Samir Abdelhafiz

---

## [Author Response · Author response to Decision Letter 1]

25 Dec 2020

Rebuttal letter

Dear Editor,

Dear Reviewers,

We are very much grateful for the Editor and reviewers’ time and willingness to
review the manuscript. We thank them for their constructive comments and
suggestions. Based on your recommendations, we addressed all the issues raised and
the corrections are included in the revised version of the manuscript.

In the following, we provide details of the changes added to the manuscript, in
respect to your valuable comments.

Reviewer #1

Thank you for making the required changes. The manuscript has improved a lot. I have
some minor comments. Kindly find the comments below 

1. Page 3, line 19: You mentioned that "however, in rare cases, patients may develop
a severe acute respiratory syndrome that requires mechanical ventilation" I think
that "rare" is not the correct term here. Maybe you can add a reference which
includes the percentage of cases that develop ARDS.

Response: Thank you for your observation. Now, we have modified the statement based
on your recommendation. 

2. Table 3, question one: The first question should be modified into "How often are
you maintaining physical distance?

Introduction 

Response: Thank you for your suggestion. We accepted the suggestion and included in
manuscript. 

3. Table 4: The p values for some parameters are not written. Please add them.

Responses: Thank you comment. Previous we included the p-values and adjusted odds
ratio (AOR) for only variables that are significant at p-value less than 0.05 in the
table of multivariable result. Now based on your recommendation, we have included p-
values and AOR for all variables in the models. 

4. Page 14, line 17: You mentioned that "Moreover, hotels, restaurants, and
cafeterias allowed services to more than three patrons at a single table" I think
you mean that that they didn't allow service to more than 3 persons at a single
table. If so, please correct the sentence.

Responses: Thank for your nice observation. It was typo error and now we corrected
it.

---

## [Editor Report · Decision Letter 2]

7 Jan 2021

Knowledge of COVID-19 and preventive behaviors among waiters working in food and
drinking establishments in Southwest Ethiopia

PONE-D-20-25994R2

Dear Dr. Asefa,

We’re pleased to inform you that your manuscript has been judged scientifically
suitable for publication and will be formally accepted for publication once it meets
all outstanding technical requirements.

Kind regards,

Jennifer A Hirst, DPhil

Academic Editor

PLOS ONE
---

## [Editor Report · Acceptance letter]

15 Jan 2021

PONE-D-20-25994R2 

Knowledge of COVID-19 and preventive behaviors among waiters working in food and
drinking establishments in Southwest Ethiopia 

Dear Dr. Asefa:

I'm pleased to inform you that your manuscript has been deemed suitable for
publication in PLOS ONE. Congratulations! Your manuscript is now with our production
department. 

Kind regards, 

on behalf of

Dr. Jennifer A. Hirst 

Academic Editor

PLOS ONE